# Aerosol–Cell Exposure System Applied to Semi-Adherent Cells for Aerosolization of Lung Surfactant and Nanoparticles Followed by High Quality RNA Extraction

**DOI:** 10.3390/nano12081362

**Published:** 2022-04-15

**Authors:** Mélanie M. Leroux, Romain Hocquel, Kevin Bourge, Boštjan Kokot, Hana Kokot, Tilen Koklič, Janez Štrancar, Yaobo Ding, Pramod Kumar, Otmar Schmid, Bertrand H. Rihn, Luc Ferrari, Olivier Joubert

**Affiliations:** 1Institut Jean Lamour, UMR CNRS 7198, Université de Lorraine, CNRS, IJL, F-54000 Nancy, France; romain.hocquel@univ-lorraine.fr (R.H.); kevin.bourge@univ-lorraine.fr (K.B.); bertrand.rihn@univ-lorraine.fr (B.H.R.); luc.ferrari@univ-lorraine.fr (L.F.); olivier.joubert@univ-lorraine.fr (O.J.); 2Jožef Stefan Institute, Department of Condensed Matter Physics, 1000 Ljubljana, Slovenia; bostjan.kokot@ijs.si (B.K.); hana.kokot@ijs.si (H.K.); tilen.koklic@ijs.si (T.K.); janez.strancar@ijs.si (J.Š.); 3Institute of Lung Health and Immunity, Helmholtz Zentrum München, German Research Center for Environmental Health, 85764 Neuherberg, Germany; dyb450303@gmail.com (Y.D.); pramod.kumar@helmholtz-muenchen.de (P.K.); otmar.schmid@helmholtz-muenchen.de (O.S.)

**Keywords:** air–liquid interface, nanoparticles, VITROCELL^®^ Cloud System, RNA extraction, macrophages, surfactant, NR8383, transcriptomic

## Abstract

Nanoparticle toxicity assessments have moved closer to physiological conditions while trying to avoid the use of animal models. An example of new in vitro exposure techniques developed is the exposure of cultured cells at the air–liquid interface (ALI), particularly in the case of respiratory airways. While the commercially available VITROCELL^®^ Cloud System has been applied for the delivery of aerosolized substances to adherent cells under ALI conditions, it has not yet been tested on lung surfactant and semi-adherent cells such as alveolar macrophages, which are playing a pivotal role in the nanoparticle-induced immune response. Objectives: In this work, we developed a comprehensive methodology for coating semi-adherent lung cells cultured at the ALI with aerosolized surfactant and subsequent dose-controlled exposure to nanoparticles (NPs). This protocol is optimized for subsequent transcriptomic studies. Methods: Semi-adherent rat alveolar macrophages NR8383 were grown at the ALI and coated with lung surfactant through nebulization using the VITROCELL^®^ Cloud 6 System before being exposed to TiO${}_{2}$ NM105 NPs. After NP exposures, RNA was extracted and its quantity and quality were measured. Results: The VITROCELL^®^ Cloud system allowed for uniform and ultrathin coating of cells with aerosolized surfactant mimicking physiological conditions in the lung. While nebulization of 57 μL of 30 mg/mL TiO${}_{2}$ and 114 μL of 15 mg/mL TiO${}_{2}$ nanoparticles yielded identical cell delivered dose, the reproducibility of dose as well as the quality of RNA extracted were better for 114 μL.

## 1. Introduction

During the recent years, the development of nanotechnologies and their many possible applications has led to their incorporation into consumer products. Indeed, in 2013, the nanotechnology Consumer Products Inventory (CPI) listed 1814 consumer products containing nanomaterials from 622 companies in 32 countries, including health and fitness products, medicine, cosmetics, food, and electronics. Predictably, this list has been growing since 2005, in all areas of production, leading to serious concerns regarding the effects of their exposure on human health [1,2,3]. Moreover, the emergence of nanotechnologies has led to the development of new research fields such as nanomedicine, which focuses on the synthesis and engineering of nanomaterials for drug delivery [4], which has led to novel treatment [5,6,7,8] and diagnostic options [9,10] and combinations thereof [11,12]. Consequently, the study of both adverse and therapeutic health effects of nanoparticles and new nanomaterials is a crucial issue for the development of safe nano-enabled products such as consumer products as well as medical products.

Generally, nanoparticle (NP) exposure is associated with an increased risk of cardiorespiratory morbidity and mortality, cancers, asthma, pulmonary diseases, fibrosis, or allergies. At the cellular scale, NPs can induce oxidative stress, inflammation, genotoxicity, membrane alterations, mitochondrial dysfunction, cell cycle arrest, and cell death [13,14,15]. Numerous studies have shown that the health effects of NPs do not depend only on their chemical and physical properties such as their size, shape, composition, surface area, or solubility, but also on the conditions of exposure including the type of exposed populations, as well as duration, doses, repetitions, and routes of exposure [16]. In the environment, the main route of exposure to nanoparticles is the respiratory route, although effects on the skin, eyes, or gastrointestinal tract need to be taken into consideration. Indeed, because of their small size and chemical properties, NPs can reach the alveoli and cross the blood–air barrier, thus infiltrating the bloodstream and other organs, for example the gastrointestinal tract, the heart, or the brain [17,18,19,20].

Over the last two decades, several attempts have been made to reduce and replace in vivo experimentation with novel in vitro models trying to mimic physiological conditions. These models tend to move closer to organ structures and, in the case of the respiratory airways, to bronchial/alveoli architecture. Among these models, lung cells cultured at the air–liquid interface (ALI) appear to be a promising technique as it offers many possibilities for mimicking parts of the respiratory tract. In the ALI culture, cells are grown on the microporous membrane of cell culture inserts, which are often placed on multi-well plates. The basolateral compartment is filled with medium and, once the cells adhere to the microporous membrane, the medium contained in the apical side of the insert is removed, establishing ALI conditions [21].

Traditional in vitro systems for nanotoxicology studies are based on the dissolution of non- or poorly-soluble particles directly in the cell culture medium, where cells are subsequently cultured [22,23]. This so-called “submerged exposure method” is not representative of actual exposure to airborne particles regarding the behavior of particles in the air [24] and implies numerous disadvantages including poor control of cell-delivered dose and interactions of the NPs with the medium [25,26]. That is why scientists and companies have developed new methods to expose cells to NPs through a gaseous phase including roller bottles, inverted cell monolayer, and smoking machines [27,28].

For inhalation toxicology, the most pertinent methodologies include ALI cell cultures and aerosol–cell exposure devices, which differ in the method used to aerosolize nanoparticles. For example, the Versatile Aerosol Concentration Enrichment System (VACES) uses a condensation process to aerosolize NPs [29]. The Electrostatic Aerosol In Vitro Exposure System (EAVES) guides aerosol particles horizontally across cell culture inserts (where cells are grown) and uses electrostatically enhanced diffusion and sedimentation for the deposition of particles on the cells [30]. The CULTEX^®^ stagnation point flow system deposits aerosolized particles via diffusion and sedimentation directly onto the cells (Cultex laboratories GmbH, Hannover, Germany) [31,32], and the Nano Aerosol Chamber In Vitro Toxicology (NACIVT) [33] uses a stagnation point flow system and an electrostatic precipitation system to deposit aerosolized NPs onto the cells. A recently developed device by Inhalation Sciences (Huddinge, Sweden), called XposeALI, uses a high-pressure aerosol generator to aerosolize dry NP powder [34] and utilizes diffusion and sedimentation for aerosol–cell deposition in a stagnation point flow setting. For continuous flow exposure, the VITROCELL^®^ Automated Exposure Station was described for testing the toxicity of combustion aerosols at the air–liquid interface [35]. Moreover, nebulization of nanoparticle suspensions into an exposure chamber with subsequent cloud settling has been introduced as ALICE and ALICE Cloud technology for dose-controlled and efficient NP–cell delivery at high dose rates by Lenz et al., 2009, and Lenz et al., 2014, respectively [36,37]. For more details on the various types of ALI exposure systems, interested readers can refer to Ehrmann et al., 2020 and Paur et al., 2011 [38,39].

The main limitation of most of the aforementioned systems is the detection and quantification of the dose of NP delivered to the site of exposure. However, it represents one of the most important criteria for the assessment of particle-related risks [16]. Indeed, most studies provide precise data on the concentration of exposure to particles, i.e., the mass of particles per volume of air (μg/m${}^{3}$) or—for cell assays immersed in vitro—per volume of cell culture medium (μg/mL), but the most relevant measure—the dose delivered to the site of exposure, e.g., lung epithelium (in vivo) or lung cell culture (in vitro)—is often overlooked. Quartz crystal microbalances (QCM) are piezoelectric biosensors that detect resonance frequency variation of quartz crystal associated with mass change on their surface. During the last decade, numerous studies have reported new possible applications of QCM in biological and medical sciences including immunosensors, biosensors, microbial detectors and innovations [40,41,42,43,44,45,46,47]. QCMs have been also described as a precise device for real-time measurement of the dose-delivered in nanotoxicology studies [26,36,48]. For these reasons, a QCM has been included in the VITROCELL^®^ Cloud System (the commercial version of the ALICE Cloud system) in order to control the real-time NP dose delivered during the Cloud exposure.

Finally, the alveolar lining fluid, which contains innate surfactant and covers the alveolar epithelium, can interfere with nanoparticles and modify their physico-chemical properties such as their agglomeration, their charge state and the type of protein corona [49,50,51]. The alveolar surfactant, which is produced by alveolar type II epithelial cells [52,53,54], is an aqueous mixture of lipids and proteins, lining the surface of pulmonary epithelial cells in the alveolar region, whose function is to maintain sufficient tension and protect the alveolar sacs from collapsing at the end of each exhalation phase. The surfactant is composed of 92% lipid mass, mostly phospholipids, and 8% protein mass. The composition of the surfactant and its amphiphilic bilayer structure makes it possible to modify the ability of external compounds to reach the pulmonary epithelium [55,56,57]. Alveolar macrophages, which are covered with alveolar surfactant and capable of neutralizing NPs in the lung by phagocytic uptake, are often considered as the most important defense mechanism against NPs deposited in the alveolar region. Recent in vitro experiments have shown that surfactant decreases the cellular uptake of silica nanoparticles by up to two orders of magnitude [58]. Since the alveolar lining fluid and the alveolar macrophages are the first and second line of defense against NPs deposited in the alveolar region, a semi-adherent alveolar macrophage cell line pre-coated with a thin film of alveolar surfactant fluid before the exposure to NPs is used in our study.

This study reports the optimization of an innovative air–liquid interface system using a cloud settling exposure system by VITROCELL^®^ Systems for pre-coating of the cells with surfactant lung lining fluid and subsequent NP exposure, combined with an accurate micrometric balance to measure cell exposure in real-time. The VITROCELL^®^ Cloud System (VITROCELL^®^ Systems GmbH, Waldkirch, Germany) [37] is a refined version of the ALICE system, which has been especially developed for aerosolized NP exposure [36]. This VITROCELL^®^ Cloud system involves a vibrating membrane nebulizer, which uses a vibrating, piezoelectrically controlled, perforated membrane to induce acoustic pressure waves that squeeze the NP suspension through the pores of the membrane. This leads to the production of a dense, uniform cloud of droplets that deposits uniformly onto the cells through a process called cloud settling, resulting in an efficient and controlled dosimetry which is monitored in real-time with a highly sensitive quartz crystal microbalance.

To sum up, in vitro studies based on ALI exposure are used more and more often in toxicology studies. However, researchers using such a strategy have to face some issues including (1) using a very low concentration of NP, (2) applying a homogeneous layer of surfactant on cells, (3) using semi-adherent cells in ALI, and (4) retrieving RNA in sufficient quantity and quality to carry out transcriptomic studies. Thus, here, we present an in-depth detailed protocol answering to these points, in which each step is validated, after the optimization of the different parameters. For a better understanding, all experimental results are featured in the present paper. It could be of use for researchers using such a device.

## 2. Materials and Methods

### 2.1. Chemicals and Reagents

Chemicals and reagents were obtained from the following sources: Dublecco’s Modified Eagle Medium (DMEM), L-glutamine, penicillin-streptomycin (PS), amphotericin B, phosphate buffer saline (PBS), and fetal bovine serum (FBS) from Sigma-Aldrich (Saint-Louis, MO, USA); Transwell^®^ polyester culture membrane from Corning (Wiesbaden, Germany). Whole porcine surfactant was a generous gift from J Perez-Gil (Faculty of Biology, Complutense University, Madrid, Spain), which includes all of the water-soluble and non-water-soluble surfactant proteins. Propan-2-ol, chloroform, absolute ethanol were bought from Carlo Erba Reagents (Val-de-Reuil, France). RNA-Solv reagent was purchased from Omega Bio-tek (Norcross, GA, USA). TiO${}_{2}$ NM105 NPs (JRC reference materials, primary NP diameter 21.5 ± 7.2 nm; secondary NP diameter (agglomerate, DLS measurement) 170 ± 0.7, BET surface area 51 m^2^/g, Zeta potential 11.1 ± 0.7 mV) were provided by JRC.

### 2.2. Cell Culture

For VITROCELL^®^ Cloud exposures, NR8383 cells (CRL2192™) are a rat alveolar macrophage cell line purchased from the American Type Culture Collection (ATCC, Manassas, VA, USA). Cells were cultured in DMEM-high glucose medium supplemented with 15% FBS, 4 mM L-glutamine, 100 U/mL penicillin, 100 g/mL streptomycin, and 0.25 μg/mL of amphotericin B at 37 °C, in a 5% CO${}_{2}$ atmosphere. Cells were grown in 75 cm^2^ flasks (Sarstedt, Nümbrecht, Germany) and then passaged and seeded on 0.4 μm pore size microporous Transwell^®^ inserts at a density of 1.5 × 10${}^{5}$ cells/mL or 3 × 10${}^{5}$ cells/mL in 1 mL of supplemented DMEM medium in the apical side of the insert and 2 mL in the basolateral side, and kept at the incubator at 37 °C, 5% CO${}_{2}$, for 24 h. Before exposure, the apical medium was removed to establish ALI.

For fluorescence microscopy, a LA-4 murine lung epithelial cell line was cultured according to ATCC guidelines: cells were seeded at 30,000 cells/cm^2^ in T75 flasks and split when they reached a 80% confluency. They were cultured in 10 mL of full cell medium (a mixture of F-12K medium (Gibco), 15% FBS (Fetal bovine serum, ATCC), 1% P/S (penicillin–streptomycin, Sigma), 1% NEAA (non-essential amino acids, Gibco)) in an incubator at 37 °C with saturated humidity and 5% CO${}_{2}$. Then, cells were seeded and observed in 35 mm μ-dishes with #1.5H glass bottom (Ibidi), cells and media covered only 1/3 of the μ-dish surface (3.5 cm^2^) due to the geometry of the μ-dish.

### 2.3. VITROCELL^®^ Cloud 6 System

The VITROCELL^®^ Cloud 6 System (VITROCELL^®^ Systems, Waldkirch, Germany) is a commercially available device aimed at exposing cells at the air–liquid interface to inhaled aerosolized toxins through cloud settling, which mimics realistic inhalation exposure scenarios of alveolar macrophages to nanoparticles in physiological conditions. The device is schematically depicted in Figure 1. The stainless-steel base module of this device comprises 5 electrically heated wells in which 13 mL of medium and the inserts are placed. The temperature in the wells is maintained at 37 °C. A sixth well contains a quartz crystal microbalance (QCM) for cell-delivered dose measurements with a resolution of 10 ng/cm^2^, at a sampling rate of 1 Hz and a manufacturer-specified zero noise level of 20 ng/cm^2^ (VITROCELL^®^ Systems, Waldkirch, Germany) that allows assessment of not only cell-delivered dose but also reproducibility and repeatability of the exposures. The QCMs used in the VITROCELL^®^ Cloud systems were well described previously in Ding et al., 2020 [48]. Briefly, the QCM incorporated in the VITROCELL^®^ Cloud 6 system has an eigenfrequency of 5 MHz, a resistance of 10 Ohm and an aerosol-exposed area of 4 cm^2^, which is close to the cell covered area in 6-well Transwell^®^ inserts (4.2 cm^2^). The upper part of the device is a 2250 cm${}^{3}$ polycarbonate removable exposure chamber pierced on the top with a hole to place the nebulizer. The Transwell^®^ inserts (Corning^®^ Transwell^®^-Clear 6-well Inserts, 10 μm thick Polyester (PET) membrane, product number 3450) used for culturing cells under ALI conditions are composed of a microporous (0.4 μm pore size) polyethylene terephthalate (PET) membrane of a surface of 4.67 cm^2^.

### 2.4. Fluorescent Microscopy for the Assessment of the Homogeneous Deposition of Surfactant

In order to compare the different methods for the deposition of surfactant on the cells, two different methods were used: nebulization or pipetting. In both cases, the cell medium was completely removed before the process. However, in some cases, the surfactant was administered immediately afterwards (leaving around 10 μm of medium on top), whereas in others the medium was left to evaporate for a few minutes before administering the surfactant (referred to as dry cells, leaving only 10 nm–100 nm layers of media on the cells).

For nebulization, 6 μL (240 μg) of the surfactant were deposited onto the pre-wetted nebulizer and were nebulized onto the entire μ-dish (surface: 3.5 cm^2^). Of these 240 μg, only ca. 7% reached the bottom surface, and ca. 2.5% (6 μg) were deposited on the cells (which are seeded only on the inner 3.5 cm^2^ of the dish due to the geometry of the dish), corresponding to approximately 10 monolayers on the cells. When pipetting, the surfactant was added by a pipette to the middle of the μ-dish (0.72 μL, 10 mg/mL), onto the medium on cells, corresponding to 10 monolayers if spread out over the whole medium-covered area of the μ-dish (3.5 cm^2^).

The surfactant used was either whole native surfactant (extract of porcine BAL, containing all surfactant proteins (SP-A, SP-B, SP-C and SP-D), kindly provided by J. Perez-Gil) or Curosurf^®^ (poractant alfa, the extract of porcine lung surfactant, consisting of 99% polar lipids and 1% SP-B and SP-C, by Chiesi). The surfactant in all experiments was labelled with STAR RED-DPPE (Abberior) using a Nlabel:Nlipid = 1:1000 labelling ratio and was dispersed in PBS to final concentration.

Delivery was performed using a pipette or a pre-wetted Aeroneb^®^ Lab Nebulizer Unit, Standard VMD (Aerogen Pro Standard VDM, Aerogen Inc., Galway, Ireland) in a tube-like setup (with the surface of the bottom being 9.6 cm^2^, see Appendix A), with an estimated rate of nebulization 5 μL/s. A tightly sealed 7 cm high cylindrical chamber with a diameter of 3 cm was adapted for live nebulization on the microscope onto a single μ-dish (see Appendix A). We estimated that 7% of initially nebulized material reached the bottom of the μ-dish (see Appendix A). The cells and surfactant were kept at 37 °C at the time of surfactant delivery to assure the surfactant was as fluid as under physiological conditions when deposited on the apical surface of the cells.

The TiO${}_{2}$ nanotubes, characterized and fluorescently labelled as described in Kokot et al., 2020 and Urbančič et al., 2018 [59,60], were dispersed in deionized H${}_{2}$O and concentrated to concentration of 33 mg/mL in a centrifuge at 21,000 rpm for 20 min (74,000×*g* RCF) (initial volume of 1 mL and concentration 1 mg/mL). Nanoparticles were nebulized either to a dry μ-dish for the panoramas or to cells with a previously nebulized layer of surfactant. The same nebulization setup as described above was used (3 μL, 33 mg/mL, corresponding to a surface area dose of nanoparticles 1:1, i.e., 1 cm^2^ of NP/per cm^2^ cells).

Different Aeroneb^®^ Lab Nebulizer Units were used for nebulizing surfactant and nanoparticles and they were thoroughly cleaned in between exposures. The images were acquired with an Abberior Instruments STED microscope equipped with a 60× water immersion objective (Olympus) and the associated software Imspector (version 16.3.11462-metadata-win64). The fluorescence was excited with two pulsed lasers at 561 and 640 nm, whereas two avalanche photodiodes at 580–625 nm and 655–720 nm, respectively, were used for fluorescence detection (filters by Semrock).

Panoramic images were combined from side-by-side recorded 0.7 × 0.7 mm large images with a 0.05 mm overlap and pixel size of 1 μm. The images were acquired over more than 1 cm using an air 10× objective (Olympus). The illumination over such a large field of view is not completely homogeneous, hence the darker edges on the images. Wolfram Mathematica was used to overlay and analyze the panoramic images. Power of excitation lasers was 5 μW.

The time series was acquired using a pixel size of 100 nm, dwell-time 10 μs, and 561 nm and 640 nm laser powers at around 5 μW, while simultaneously imaging using two-photon excitation with wavelength at 950 nm, dwell-time 400 μs, and power at around 10 mW (the two-photon excitation and the entire time-lapse are shown in Appendix A). The surfactant structure following nebulization of 100 monolayers (2.4 mg of Curosurf) was acquired using two-photon excitation at 950 nm, pixel size of 250 nm, and dwell-time 8 ns.

### 2.5. Pre-Coating of ALI Cells with Surfactant

The freeze-dried whole surfactant powder extract from pig lungs was dissolved in 100 mL ultrapure water at room temperature, yielding a concentration of 70 mg/mL, and sonicated 30 s at 37 °C in ultrasonic cleaner. It was then diluted in PBS to reach a working concentration of 40 mg/mL, sonicated 30 s at 37 °C in ultrasonic cleaner, and left at 37 °C in the incubator for 1 h. After the removal of the apical medium on the Transwell^®^ inserts, 61 μL of the surfactant solution were nebulized with the VITROCELL^®^ Cloud 6 System allowing for cloud settling time. In the initial protocol, the nebulization of surfactant was performed just before the NP nebulization, while, for the current protocol, surfactant nebulization was conducted for all samples together (control and NP exposed). The inserts were then placed back into the standard multiwell plates and were incubated at 37 °C, 5% CO${}_{2}$, until a second nebulization (NP exposure, or—for controls—water exposure) was performed with the VITROCELL^®^Cloud system.

### 2.6. NP Exposure Protocol

TiO${}_{2}$ NM105 NPs were suspended in 5 mL of ultrapure water at a concentration of 30 mg/mL and sonicated for 10 min in an ultrasonic cleaner (VWR International, Radnor, PA, USA) just before exposure with the VITROCELL^®^ Cloud 6 System. As described above, NR8383 cells were seeded on the apical side of Transwell^®^ inserts at a concentration of 1.5 × 10${}^{5}$ cells/mL (initial protocol) or 3 × 10${}^{5}$ cells/mL (current protocol) in 1 mL of supplemented DMEM medium for 24 h. In each well of the VITROCELL^®^ Cloud 6 System, 13 mL of medium were added and inserts were placed over the wells to set up the air–liquid configuration, 61 μL of surfactant were then nebulized on the inserts after removal of the apical medium. Subsequently, the cells were exposed to 57 μL of aerosolized nanoparticle suspension or an equivalent volume of DMEM medium or ultrapure water (controls). After exposure, the cells were placed in the incubator for 4 h and then recovered to extract their RNA. In this study, the nebulizer was rinsed between each exposure by nebulizing 1 mL of ultrapure water (Ultrapure (Type1) Water Simplicity^®^, filter Millipak^®^ 20 Millipore 0.22 μm) followed by 1 mL of PBS 1X. The nebulizer was also sonicated from time to time if it was necessary to address the clogging of the mesh.

The nebulization of the nanoparticles and surfactant suspensions was performed with an Aeroneb^®^ Lab Micropump Nebulizer (Aerogen Inc., Galway, Ireland). The nebulizer system is composed of the Aeroneb^®^ Lab Nebulizer Unit, Standard VMD and an Aeroneb^®^ Lab Control Module (Aerogen Inc., Galway, Ireland) (Figure 1). The nebulization relies on a perforated piezoelectrically controlled vibrating mesh to generate acoustic pressure waves releasing liquid droplets at a high frequency (128 kHz). This nebulizer can be used in vitro, in vivo, or in clinical settings [61,62] and is constituted of a palladium mesh pierced with ca. 1000 holes, which vibrates at 128,000 times per second, releasing between 0.3 and 0.8 mL of liquid per minute through the holes, resulting in a stream of droplets precisely controlled for size by the diameter of the apertures (4–6 μm droplet diameter for the Standard VDM nebulizer) [62]. For more information on the Aeroneb^®^ nebulizer system, please refer to the manufacturer website (Aerogen Technology).

### 2.7. RNA Extraction

Here we present the first step of transcriptomic study [63,64], i.e., RNA extraction. Excellent quality and sufficient quantity of RNA is a prerequisite for transcriptomic assay. Indeed, the concentration of total RNA should be more than 40 ng/μL and the quality is defined by the ratios of optical densities (OD) 260/230 and 260/280 which must be between 1.8 and 2.2.

A total of 4 h after exposures, the culture media was removed from the basolateral and apical compartments of the inserts. Cells were lysed by flushing 1 mL of RNA-Solv reagent directly onto the inserts membrane and then delicately scrapped from the membrane with a pipet tip to detach every single cell. All technical replicates (TR) originating from the same biological replicate (BR) were pooled. Samples were then stored at −80 °C overnight. The next day, 200 μL of chloroform per tube were added and samples were centrifuged at 12,000× *g* for 15 min at 4 °C. An amount of 175 μL of propan-2-ol was added to 350 μL of supernatant and centrifugation was again proceeded at 12,000× *g* for 20 min. Pellets were then subjected to two successive ethanol washing steps and then dried at 60 °C for 10 min to remove the excess of ethanol. Finally, dried pellets were diluted in 25 μL of RNase-free water. RNA quantity and purity were assessed using a BioSpec-nano spectrophotometer (Shimadzu, Marne-la-Vallée, France).

## 3. Results

### 3.1. A Uniform Deposition of Surfactant by Aerosolization with the Nebulizer

#### 3.1.1. Reproducible Deposition Assessment by QCM Measurements

The thickness of the alveolar lung lining fluid in rat lungs ranges between 0.1 μm and 0.9 μm. The alveolar lining layer appears continuous, submerging epithelial cell microvilli and intercellular junctional ridges, and serves to smoothen the alveolar air–liquid interface in inflated lungs [65]. The standard operating procedure of the VITROCELL^®^ Cloud System provided by VITROCELL^®^ Systems recommends nebulization of 200 μL of liquid, which results in ca. 5.6 μL of liquid aerosol deposited per six-well (4.5 cm^2^) Transwell^®^ insert (the values can vary depending on nebulizer performance). This corresponds to a ca. 12 μm thin liquid film, which evaporates within a few seconds from the QCM after opening the exposure chamber. It has been shown that a thin aqueous layer like this has no adverse effects on cellular response [36]. To adapt the thickness of the in vitro surfactant coating to the in vivo conditions, we only nebulized 61 μL of surfactant solution (at 40 μg/μL) or 57 μL of nanoparticle suspension (in our initial protocol, against 114 μL of NP suspension in the current protocol). It reflects the minimum volume of nebulized liquid (ca. 60 μL) to still allow for relatively uniform cloud mixing in the VITROCELL^®^ Cloud System, which is a prerequisite for reliable operation. As the surface of the VITROCELL^®^ Cloud chamber is 143 cm^2^, the surfactant mass initially nebulized corresponds to 76.5 μg/insert and 1.9 μL/insert (17 μg/cm^2^ and 0.42 μL/cm^2^), resulting (theoretically) in a 4 μm aqueous layer. Although slightly higher than in vivo conditions (which, as a reminder, ranges between 0.1 and 0.9 µm), this is the least we can obtain within the limitation of nebulization of at least 60 μL for uniformity of aerosol deposition in the VITROCELL^®^ Cloud 6 system. However, the QCM has not provided reliable measurements of deposited surfactant mass as the reported QCM signal showed a continuous growth rather than the typically observed peak value after a few minutes and subsequent adjustment to a constant asymptotic value. This could be related to the fact that alveolar surfactant contains a high concentration of surface-active proteins, which inhibits evaporation of the aqueous phase. Indeed, in the presence of water or any other liquid, the QCM signal is more representative of the viscosity of the liquid than the deposited mass [36,48]. Nevertheless, the QCM signal demonstrates the high degree of reproducibility of surfactant deposition on a qualitative level, but the reported mass dose does not reflect surfactant mass (Figure 2). 

#### 3.1.2. Homogeneous Distribution Assessment by Fluorescence Microscopy

In parallel, the homogeneous distribution of nebulized surfactant over the cells was assessed by fluorescence microscopy. The deposition of surfactant on the cells does not depend on the cell type. Therefore, the dosimetry data reported here for LA-4 cells also apply to any other cell type, including the NR8383 cells. Surfactant was labelled with STAR RED-DPPE fluorescent dye and nebulized or pipetted on a monoculture of LA-4 murine lung epithelial cells in ALI conditions. Fluorescence top-view panoramic scanning was then used to observe the surfactant deposition (Figure 3). As confirmed using a fluorescence top-view panoramic scan over 1 cm (one half of the sample diameter), the nebulization of surfactant (Figure 3A(iii),B(iii)) covers the entire sample with a homogeneous surfactant layer. In contrast, an alternative method of surfactant application, i.e., pipetting the surfactant directly onto the cells at the ALI (Figure 3A(i),B(i)) or onto cells slightly covered with cell medium (Figure 3A(ii),B(ii)), does not provide a uniform surfactant deposition. Moreover, the pipetted surfactant does not spread over the whole sample, leaving the majority of the cells without surfactant. The fluorescence panoramic scan of nebulized nanomaterial confirms that nebulization also delivers the nanomaterial uniformly over the entire sample (Figure 3C). As shown in Figure 3D, after nebulization onto the cells, surfactant formed a homogeneous layer on the air–liquid interface, with the remainder of the surfactant-forming structures in the underlying aqueous cell medium (Figure 3E). These results confirm that, when subsequent nebulization of surfactant and nanoparticles is performed, the resulting sample is uniform both in terms of surfactant and the cell-delivered dose of nanomaterial (Figure 3F). This is crucial for replicating the physiological conditions.

### 3.2. Improvement of the Standard Initial Protocol

#### 3.2.1. Standard Initial Protocol for NP Nebulization in VITROCELL^®^ Cloud 6 System

The initial exposure protocol that we used was set up as follows: 40 inserts were seeded with NR8383 cells out of four biological replicates (BR, 10 inserts per BR, 150,000 cells/insert) and incubated 24 h at 37 °C, 5% CO${}_{2}$. After the removal of their supernatant and the exposure to surfactant, establishing ALI, half of the inserts (five inserts for each BR, representing five technical replicates (TR) for each BR exposed) were exposed to ultrapure water as control and the other half to NM105 NP (Figure 4). Cells were then put back into the incubator for 4 h and their RNA was finally extracted. Running this very first protocol, we encountered a few issues and realized it had a few flaws that could be corrected to be more biologically rigorous and to optimize RNA yield and quality.

#### 3.2.2. Current Improved Protocol for NP Nebulization in VITROCELL^®^ Cloud 6 System

The current exposure protocol makes use of five BR of which eight TR were generated. For each BR, half of the inserts (four inserts for each BR, representing four TR) were exposed to ultrapure water as a control and the other half to NM105 NP (Figure 5). A total of 300,000 cells were seeded in 1 mL of complete growth medium by inserts and allowed to incubate for 24 h at 37 °C, 5% CO${}_{2}$. Before exposure, the apical medium contained in the inserts was removed to establish the ALI configuration, 61 μL of surfactant were then nebulized over the inserts, and the inserts were placed back into the incubator. Once each insert was exposed to surfactant, 114 μL of TiO${}_{2}$ NP at a concentration of 15 mg/mL were nebulized over the cells. cells where then incubated for 4 h and the TR of each BR were pooled together for RNA extraction.

### 3.3. Troubleshooting Flaws of the Initial Protocol

#### 3.3.1. Preventing Rapid Drying of NPs on the Nebulizer Mesh by Modifying the Volume of Nebulized Liquid

The first issue encountered with these exposure parameters was that the TiO${}_{2}$ NP suspension tends to rapidly dry out on the mesh of the nebulizer, which is likely a result of partial clogging of the pores of the mesh, making the TiO${}_{2}$ NP depositions variable in time and quantity (Table 1; Figure 6A). For a concentration of 30 mg/mL of NM105 TiO${}_{2}$ NP suspended in 57 μL of ultrapure water, we observed a mass deposition ranging from 2.1 to 7.4 μg/cm^2^ and the nebulization time needed for a stable QCM signal varied from 6 min to 24 min (Table 1). To address these high variations, we increased the volume to 114 μL and reduced the concentration to 15 mg/mL (current protocol) by doubling the nebulized volume and reducing the TiO${}_{2}$ NP concentration by two, so the same dose was present in the nebulizer as for the initial condition. This was aimed at maintaining the cell-delivered mass dose but reducing the range of variations in mass dose and in exposure time (Table 1; Figure 6B). This improved the conditions from 5.1 +/− 2.7 μg/cm^2^ to 5.4 +/− 0.8 μg/cm^2^ and from 10.5 +/− 9.0 min to 5 min for 30 mg/mL and 15 mg/mL, respectively. Hence, the mass deposition was less variable, which can also be seen from the QCM signals (Table 1). It is noteworthy that, unlike previous studies with the VITROCELL^®^ Cloud 6 system, we did not lift the exposure chamber after 3–5 min of cloud settling, since we did not want to disturb the potentially slower cloud settling of highly concentrated surfactant and TiO${}_{2}$ NM105 aerosol. This prolongs the time for evaporation of the liquid layer on the QCM and hence extends the time for reaching an asymptotic QCM value as compared to previous studies [36,48].

#### 3.3.2. Loss of Cells and Poor RNA Quality

NR8383 cells are a semi-adherent cell line, indeed half of the cell population adheres to the membrane of the insert whereas the other half stays in suspension. The issue is that the supernatant has to be removed before exposing cells to surfactant to reach ALI, leading to a loss of about 50% of cells (75,000 with the initial protocol). This issue sometimes resulted in a too-small amount of RNA and the loss of the pellet while performing washing steps during the RNA extraction. The low number of cells also forced us to retrieve a bigger fraction of the aqueous phase after adding chloroform, increasing the risk of protein or salt contaminations (optical density (OD) ratios should be between 1.8 and 2.2) (Table 2). To fix this issue, we doubled the cell density in the inserts, going from 150,000 cells/insert to 300,000 cells/insert and this led to an improvement in reproducibility and RNA quality. Mean RNA quantity went, indeed, from 83.61 ± 73.27 ng/μL to 50.46 ± 26.34 ng/μL, and OD${}_{260/230}$ ratio increased from 1.33 ± 0.70 to 2.16 ± 0.59 (Table 2).

#### 3.3.3. Increasing the Reproducibility and Repeatability

TiO${}_{2}$ NP mass deposition variability was also addressed by a change in the organization of the exposures. With the initial protocol, we used four biological replicates (BR) divided these into five technical replicates (TR) (Figure 4). Each TR of a BR was exposed to TiO${}_{2}$ NPs at the same time in the VITROCELL^®^ Cloud chamber, during the same nebulization (Figure 4). This initial protocol was less rigorous as it implied that the four BR would be exposed during different nebulizations and, consequently, potentially to a different dose of TiO${}_{2}$ NPs because of the variability of mass deposition between each nebulization, introducing a bias in the comparison of the effects of TiO${}_{2}$ NP on each BR. To reduce this bias, we worked with four TR from five BR. One insert (TR) of each BR was placed in a well of the VITROCELL^®^ Cloud System and exposed to TiO${}_{2}$ NP Nebulization. This was realized for each of the four TR, and the four TR of each BR were pooled to proceed to the RNA extraction (Figure 5). This allowed us to smoothen the nebulization variation bias on every BR, as each would receive the very same dose of TiO${}_{2}$ NP; thus, the mass deposition variations would be the same for every BR (Figure 7).

## 4. Discussion

Air–liquid exposure systems do represent closer-to-life scenarios in comparison to the traditional and currently most widely used submerged exposure methods, and this is not limited only to the Cloud systems. One of the main disadvantages of most aerosol–cell exposure devices (VACES, EAVES, CULTEX, NACIVT) is the lack of mass deposition control and—maybe more importantly—low dose rate, which often requires days to weeks of exposure time to observe any biological response in vitro cell culture models [39]. The choice of using VITROCELL^®^ Cloud was based on short exposure times (a few minutes) and accurate control over the deposited NP mass dose on the cells using the quartz crystal microbalance. The latter enables the possibility to report the deposition variations on the results and to normalize the effects of NP dose on cell viability and gene expression. This control allows even more precise interpretations of the results. Moreover, the VITROCELL^®^ Cloud system allows for surfactant coating of lung cells by nebulization before NP exposure, which is particularly relevant for cell models without innate surfactant secretion (e.g., alveolar macrophages). The use of surfactant provides the environment, closer to the lung physiological conditions, with controlled surface ratios between cells, surfactant, and nanomaterial.

It has been shown that surfactant can alter the physico-chemical properties of NPs and hence modulate their toxicity [49,50,51]. Since the nebulized surfactant completely covers the entire sample surface, the underlying cells are never directly exposed to the nanoparticles. Thus, the here-described method prevents possible artefacts arising from the direct interaction of the nanoparticles with the cells without surfactant. Moreover, nebulization of nanoparticles also prevents possible unrealistically high local doses of nanoparticles, the so called “hot spots” as described for in vivo experiments when instillation is used instead of inhalation [66]. Therefore, the nebulization of surfactant helps to produce results that tend to be closer to reality and are therefore more predictive for human health outcomes, which may ultimately lead to more reliable regulatory control of exposure to NPs.

It is important to consider that our problems with the initial protocol are probably a combination of several issues. Notably, it can be due to a partial clogging of the nebulizer pores for high NP concentrations (as used here, 30 μg/μL), which then reduces the nebulizer output rate, enhances the nebulization time and then leads to partial liquid evaporation. Di Cristo et al. have published an interesting paper, with doses close to those we used, comparing the deposition efficiency of low volume (30 μL) of two TiO${}_{2}$ NPs (NM100 and NM101), at lower concentration (<100 μg/mL) nebulized with the Aeroneb^®^ Pro Nebulizer Sytem directly upon the insert (while we nebulized 57 μL at 30 mg/mL, or 114 μL at 15 mg/mL of NM105 TiO${}_{2}$ NPs with Aeroneb^®^ Lab Standard VMD Nebulizer, in VITROCELL^®^ Cloud System). They showed that the deposition of NP depends on the type of NP. Indeed, the deposition of NM-100 was homogeneous, whereas NM-101 formed small and large aggregates when nebulized. For NM-100, the doses delivered by aerosol were very similar to the initial concentration deposited in the nebulizer (90% of deposition efficiency, efficiency, i.e., 90% of invested dose deposited on the bottom of the VITROCELL^®^ Cloud system), whereas the deposition efficiency was very low for NM-101 (values below 50%). Such an observation can be associated with the formation of NP aggregates in the suspension that could remain trapped erratically on the nebulizer mesh. Indeed, the dispersion of NPs during nebulization is strongly dependent on the aggregation phenomenon occurring before and during the biological experiments [67].

Moreover, using the same nebulizer for surfactant coating and then for NP exposure increased the risk of clogging the mesh. Indeed, in the initial protocol, we nebulized back-to-back surfactant and NP in each exposition, and we observed a frequent clogging of the mesh and variation of the NP mass deposition, even if the nebulizer was cleaned between each nebulization. The clogging probably occurs because lipids stick NPs together acting like a cross-linker. We observed that labelled surfactant stuck to TiO${}_{2}$ NTs after just the TiO${}_{2}$ NTs were nebulized subsequent to surfactant nebulization (no measurements available), without in-between cleaning. This was diminished when we exposed for the first time all the 40 wells to the surfactant, and for the second time to the NP after cleaning and sonicating the nebulizer (current protocol). Hence, a rigorous cleaning procedure for the nebulizer is essential for the reliable operation of the VITROCELL^®^ Cloud system. If possible, different nebulizers should be used if two or more substances are to be nebulized back-to-back, since this alleviates the cleaning requirements and prevents possible cross-contaminations of the nebulized entities. However, even if clogging occurs, the fact that the actual dose delivered to the cells is measured with the QCM during each exposure allows for reliable dose–response curves independent of the degree of clogging.

During aerosol–cell exposure experiments, aerosol deposition is influenced by several factors, including the aerosol-generating system, aerosol characteristics (particle size, shape, density, etc.) and the inhalation pattern (flow rate and volume) [67,68]. The VITROCELL^®^ Cloud system is unique in the sense that it uses air-less (no flow rate) cloud dynamics (not single aerosol dynamics) for rapid and spatially uniform dose delivery leveraging cloud sedimentation, which only depends on a high enough aerosol/liquid volume per air volume (cloud density) [37]. Under those conditions, aerosol–cell delivery is independent of aerosol size, shape and density provided the cloud density is high enough to marginalize single aerosol deposition mechanisms as compared to cloud settling. For the conditions of the VITROCELL^®^ Cloud system, 200 μL of nebulized liquid ensures high enough liquid content of the air in the exposure chamber to ensure cloud rather than single aerosol settling conditions [37]. Since this may not be the case, if too small a liquid volume is nebulized, it is recommended to operate the VITROCELL^®^ Cloud 6 system with 200 μL. However, we have shown here that 110 μL can also be sufficient for reproducible NP delivery. In addition, Ding et al. [48] recently demonstrated that the QCM signal corresponds to mass only if the deposited NP layer is perfectly coupled to the quartz crystal. In the presence of water (even more if there is a high volume nebulized), there is viscoelastic decoupling of the NPs from the quartz and the signal is more representative of the viscosity of the deposited NP suspension layer than of NP mass. Therefore, directly after the NPs have settled onto the QCM, the top part of the exposure chamber is removed, which allows for rapid drying of the NP layer on the QCM.

One might raise the issues on realistic dosimetry and exposure routes, e.g., whether the dose is too high or delivered too fast with the VITROCELL^®^ Cloud 6 system as compared to ambient or occupational exposure scenarios. Moreover, in real-life exposure scenarios nanoparticles do not enter human lungs as a liquid droplet suspension but rather in dry form, and the NP crosses the entire respiratory tract before reaching the lung cells. However, upon deposition onto the lung lining fluid, all initially dry NPs will make contact with the liquid phase of the lining fluid. After all, nebulizers are currently the most developed system for aerosol lung delivery used in research experiments as well as in the clinic [62,69]. Dubus et al. have demonstrated that such a nebulizer (Aeroneb^®^ Pro Nebulizer) provided 12% to 14% dose deposition in macaque lung, i.e., a 25-fold greater deposition than another nebulizer (MistyNeb 0.5%); this information is useful to compare the dose delivery in ALI exposure cell-monolayer and in vivo lung exposure.

Considering the bulk of studies about TiO${}_{2}$ nanoparticle (NP) toxicity, there are few publications comparing in vitro and in vivo exposures, and even fewer comparing air-–liquid interface exposure (ALI) with other in vitro and in vivo exposures. Therefore, after ALI cell exposure to TiO${}_{2}$ NP and RNA extractions, described in this article, we aimed to identify early specific markers of lung exposure to TiO${}_{2}$ NP, by a large study of genic expression modifications, mainly transcriptomics, and comparing classical submerged in vitro, ALI and in vivo exposures [63,64]. Knowing that nanoparticles can reach the alveoli [24], NR8383 cells are an appropriate model because they are alveolar macrophage precursors, which are the first implicated cells in the alveolar clearance of nanoparticles [70,71,72]. Moreover, this cell line has already been studied and validated as a model in the field of nanotoxicology and is relevant for its immune functions [73,74,75].

Nevertheless, our methodology could be improved by using ALI co-cultures of cell lines instead of monocultures, to be even closer to physiological conditions, considering the interactions that would exist between the different cell types. The first ALI co-culture was described in the 1980s with the growth of endothelial and epithelial cells on a cellulose membrane [76]. The simplest and most used ALI model is the cell monoculture (with cell lines or primary cells) and this model was largely used for studying transepithelial electrical resistance (TEER) [77,78,79,80].

Recently, Kletting showed that by cultivating differentiated m0 macrophages (THP-1) in the presence of alveolar epithelial cells (hAELVi) at the air–liquid interface, an extracellular matrix was secreted at the surface and their transport study showed a functional air–blood diffusion barrier phenotype [81]. Such a phenotype would greatly serve our pulmonary NP toxicity study, as it would mimic even more accurately the pulmonary physiological conditions. Another study conducted by Klein et al. [82] highlighted that SiO${}_{2}$ NP exposures in ALI conditions induced a higher ROS generation in monocultures than co-cultures (macrophages + endothelial cells + bronchial epithelial cells + mast cells). It showed that the effects encountered in monocultures tend to be over evaluated compared to results obtained when culture conditions move closer to physiological conditions [82]. Currently, the best way to replicate the pulmonary environment in ALI is to set up co-cultures or 3D cultures [83]. Finally, some aerosol cell exposure systems, recently developed, include continuous airflow (VITROCELL^®^ Automated Exposure Station) [35,38,39] and even cyclic stretch, to simulate in vitro the mechano-elastic stimulation by continuous inhalation and exhalation [84,85,86].

## 5. Conclusions

The goal of this work was to refine an ALI pre-established protocol. As far as we know, such a protocol has not been previously published. Therefore, after addressing each crucial step, (i.e., the use of very low concentrations of NP, the deposition of a homogeneous layer of surfactant, the specific use of semi-adherent cells, and finally the enhancement of the extraction yield of RNA to carry out transcriptomic studies), we present here for the first time a robust almost ready-to-use protocol. This one will be helpful for nanotoxicologists interested in developing such methodologies. This protocol is not frozen, and researchers can feel free to suggest improvements.

Finally, we do not recommend replacing the existing standard operating procedure [37], but we present four optimization points when exposing semi-adherent cells to NPs under ALI conditions with the VITROCELL^®^ Cloud 6 system (1); if low volumes and high concentrations are nebulized (2); when two sequential aerosol exposures are performed back-to-back (here: lung surfactant and NPs in 61 μL and 114 μL, respectively) (3); and how to arrange the wells to reduce the variability between the biological replicates (BR) (4):(1)When using semi adherent cells, the number of cells seeded on the insert has to be increased (as compared to submerged culture conditions) to ensure that enough cells stay on the membrane after discarding the apical medium to reach a satisfying RNA yield.(2)To avoid any clogging of the mesh of the nebulizer and to obtain a reproducible cell-delivered dose, the manufacturer’s recommendations for cleaning of the nebulizer should be adhered to and the TiO${}_{2}$ NP concentration has to be decreased by increasing the nebulized suspension volume (from 57 μL of 30 mg/mL to 114 μL of 15 mg/mL in this case for TiO${}_{2}$ NM105 NP). This will prevent the NP suspension from both drying out on the nebulizer mesh and alteration of the NP deposition on the cells due to low cloud density in the VITROCELL^®^ Cloud exposure chamber.(3)When two sequential aerosol exposures are performed back-to-back, when applicable, we suggest using a different nebulizer for each liquid nebulized. If the same nebulizer is used, we suggest nebulizing all replicas with the first type of aerosol and then with the second, and cleaning the nebulizer between each sequence of nebulization thoroughly as recommended by the manufacturer.(4)If it is necessary to proceed to several expositions, we recommend including each BR in each exposure session (divided into technical replicates (TR), one TR per exposition). By doing this and subsequent pooling of the TRs of each BR, the variability in deposited dose amongst expositions will be spread over every TR in the same manner for each BR.

After the optimization of this protocol, we were able to obtain a high quality and quantity of total RNA extraction from this VITROCELL^®^ Cloud exposition and we performed a complete transcriptomic study [64] including a comparison of in vivo and air–liquid interface and submerged in vitro results for TiO${}_{2}$ NM105 NP exposure [63].

The optimization of the existing protocols is essential when moving to new or different experimental conditions (e.g., in our case of semi-adherent cell cultures, surfactant and low doses of NPs). The method that we employed to optimize/refine the previously established protocol could efficiently be transposed to 3D-cultures [87,88,89,90] or culture of organoids [91,92,93,94,95], which represent today the in vitro models closest to physiological conditions, considering the interactions existing between the different cell types. Furthermore, the approach that we used could serve as a basis for further developments which, in the long term, could significantly reduce the need of animal studies for toxicology research.

## Figures and Tables

**Figure 1 nanomaterials-12-01362-f001:**
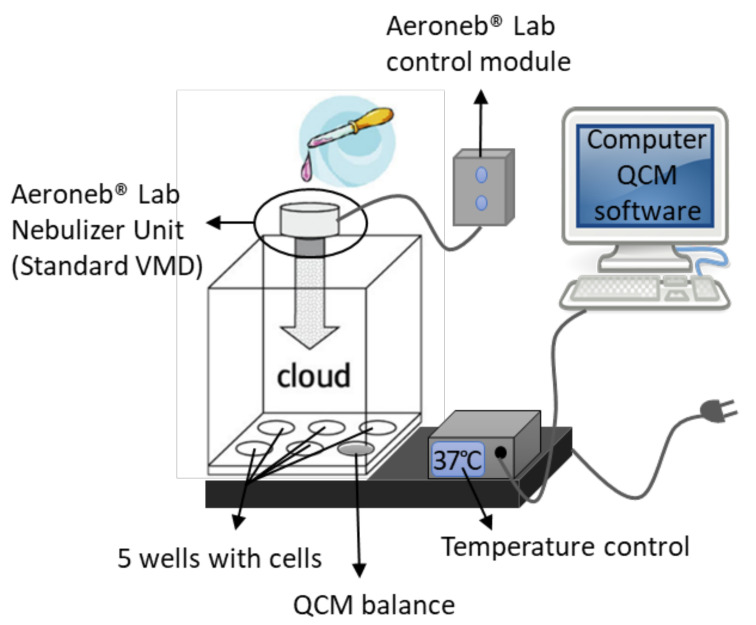
Schematic of the VITROCELL^®^ 6 Cloud device. The stainless-steel base module of this device comprises 5 electrically heated wells in which 13 mL of medium and the inserts are placed. The temperature in the wells is maintained at 37 °C. A sixth well contains a quartz crystal microbalance (QCM) for cell-delivered dose measurements with a resolution of 10 ng/cm^2^, at a sampling rate of 1 Hz and a manufacturer-specified zero noise level of 20 ng/cm^2^ (VITROCELL^®^ Systems, Waldkirch, Germany), measurements are analyzed and visualized using the computer QCM software. The upper part of the device is a 2250 cm${}^{3}$ polycarbonate removable exposure chamber pierced on the top with a hole to place the nebulizer. The nebulizer system is composed of the Aeroneb^®^ Lab Nebulizer Unit, Standard VMD and a Aeroneb^®^ Lab Control Module (Aerogen Inc., Galway, Ireland). The nebulization relies on a perforated piezoelectrically controlled vibrating mesh to generate acoustic pressure waves releasing liquid droplets at a high frequency (128 kHz). This nebulizer is constituted of a palladium mesh pierced with ca. 1000 holes, releasing between 0.3 and 0.8 mL of liquid per minute through the holes, resulting in a stream of droplets precisely controlled for size by the diameter of the apertures (4–6 μm droplet diameter for the Standard VDM nebulizer).

**Figure 2 nanomaterials-12-01362-f002:**
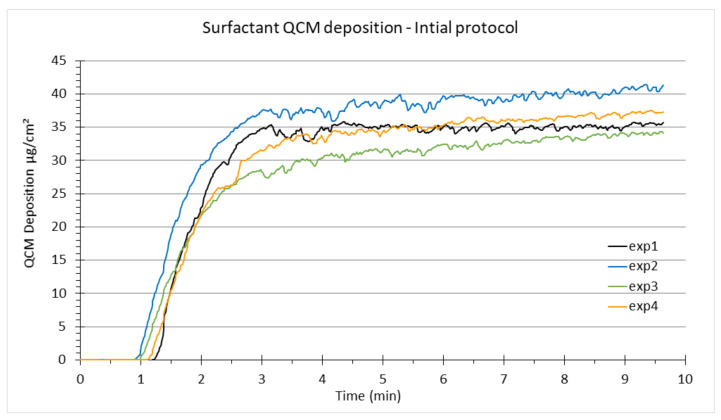
QCM curves showing the cloud deposition of surfactant (qualitatively) in the VITROCELL^®^ Cloud System. This Figure illustrates the reproducibility of the QCM response for independent exposures under the same conditions. An amount of 61 µL of whole porcine surfactant solution at 40 mg/mL was nebulized, and the VITROCELL^®^ Cloud temperature was maintained at 37 °C. The nebulization starts between 30 s and 1 min after the start of data acquisition. Depicted are the four independent exposures performed in the present study before NP nebulization.

**Figure 3 nanomaterials-12-01362-f003:**
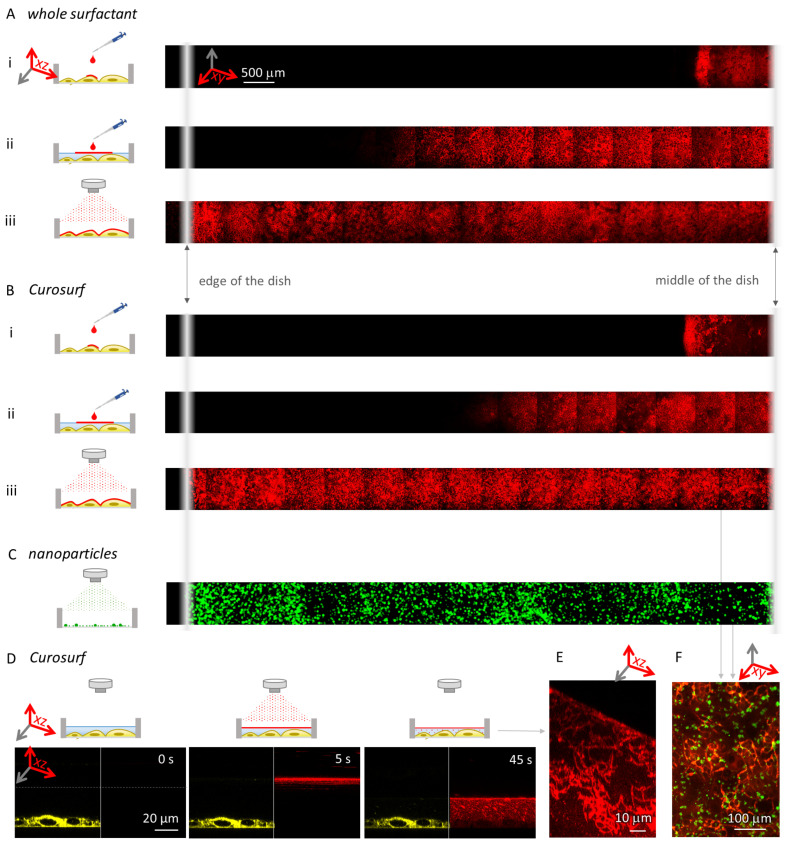
The deposition of nebulized and pipetted surfactant. A total of 10 monolayers of fluorescently labelled (**A**) whole surfactant and (**B**) Curosurf were administered onto cells using different methods. As shown in the top-view panoramas, in both cases (**i**) surfactant, pipetted onto cells with removed medium (simulating ALI conditions), spreads only over a fraction of the sample, (**ii**) surfactant pipetted onto cells slightly covered with cell medium spreads further, but still does not cover the entire sample, and (**iii**) nebulized surfactant evenly covers the entire sample (fluorescence intensities are reported in Appendix A). Note that the intensities are not comparable between measurements, and the dark vertical lines are the consequence of uneven illumination. (**C**) Nebulization of fluorescently labelled TiO${}_{2}$ nanotubes (green) evenly deposits the material over the sample—shown here in a top-view panorama for nebulization directly onto a glass surface. Note that, due to the low signal, the image was filtered using a Gauss filter, and the intensity was scaled from 0 to 3 counts. (**D**) A time series of nebulization of 10 monolayers of Curosurf (red, fluorescently labelled with STAR RED-DPPE) onto lung alveolar cells (green, labelled with CellMask Orange), (see Appendix A for complete time series). (**E**) A side-view of the surfactant structure just below the surface after nebulizing 100 monolayers of surfactant (red) onto submerged cells. The surface of the medium is tilted due to capillary effects near the edge of the dish. (**F**) A fluorescence top-view micrograph of 10 monolayers of nebulized surfactant (red) and subsequently nebulized 1:1 surface dose of nanomaterial (green) onto lung alveolar cells.

**Figure 4 nanomaterials-12-01362-f004:**
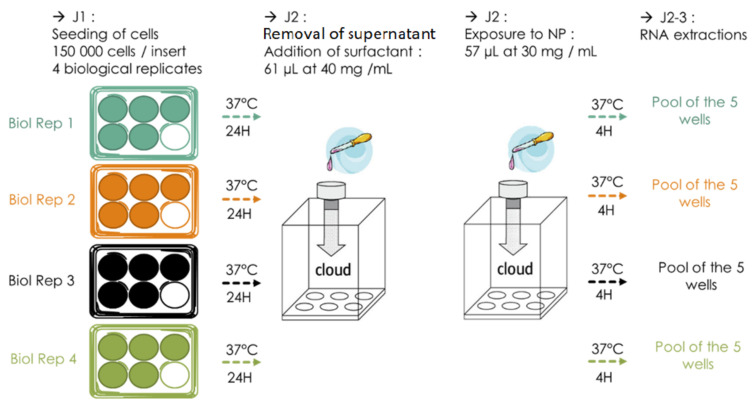
Initial exposure protocol. For each 4 biological replicates (BR), NR8383 were seeded in 5 wells, in Transwell^®^ inserts. The next day, for each BR, the inserts containing the cells were exposed, by nebulization, back-to-back with 61 µL of whole porcine surfactant at 40 mg/mL, and with 57 µL of TiO${}_{2}$ NP at 30 mg/mL. After exposures, the Transwell^®^ inserts containing the cells were kept at 37 °C for 4 h, and the 5 wells (representing technical replicates, TR) of each BR were pooled for RNA extractions.

**Figure 5 nanomaterials-12-01362-f005:**
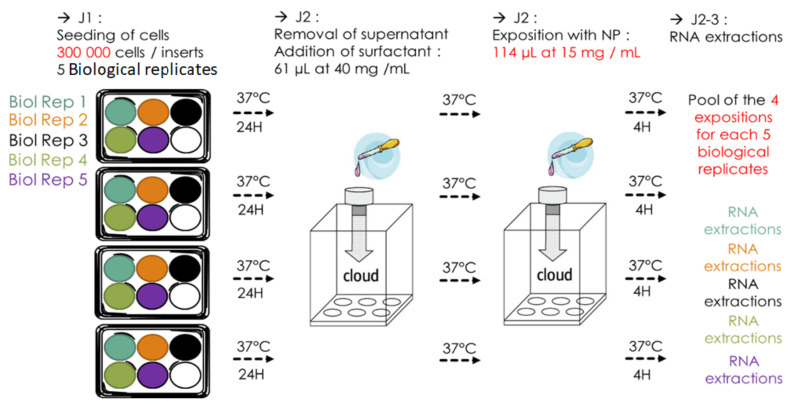
Current exposure protocol. For each 5 biological replicates (BR), NR8383 were seeded in 4 wells, in Transwell^®^ inserts, distributed in 4 6-well plates. The next day, for all 6-well plates, the inserts containing the cells were exposed (representing one technical replicate (TR) of each BR), by nebulization, first to 61 µL of whole porcine surfactant at 40 mg/mL, and then to 57 µL of TiO${}_{2}$ NP at 30 mg/mL. After exposures, the Transwell^®^ inserts containing the cells was kept at 37 °C for 4 h, and the 4 wells (TR) of each BR were pooled for RNA extractions.

**Figure 6 nanomaterials-12-01362-f006:**
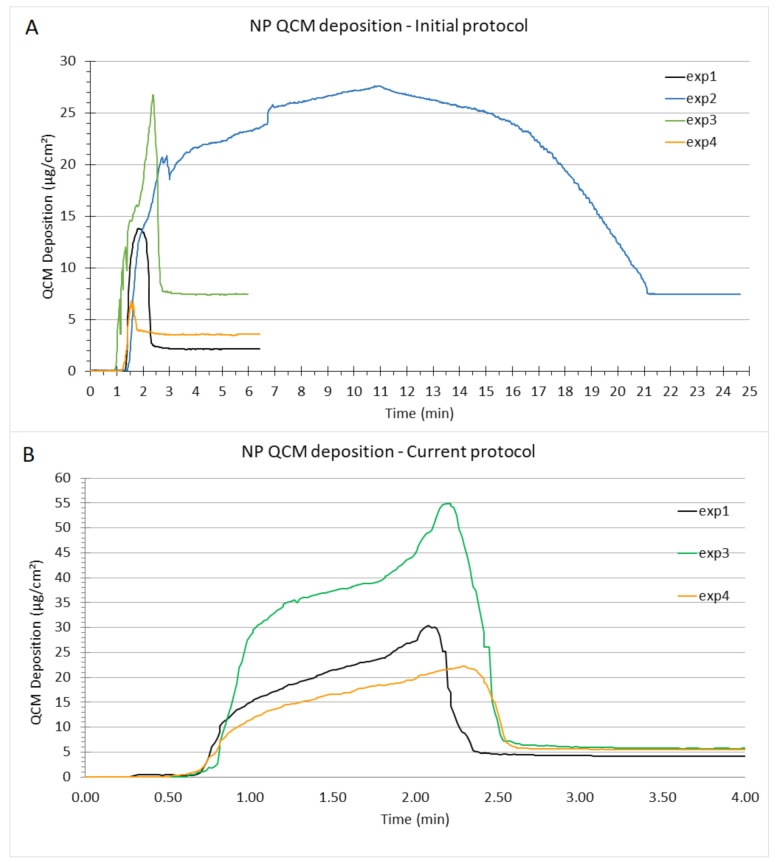
QCM curves showing in the first part the dynamics of cloud deposition in the VITROCELL^®^ Cloud System followed by an asymptotic (constant) value, which represents the mass of the deposited TiO${}_{2}$ NPs. The nebulization starts between 30 s and 1 min after the start of data acquisition. Depicted are (**A**) the four NP exposures following the initial protocol (57 µL of TiO${}_{2}$ NP at 30 mg/mL); (**B**) three exposures following the current protocol (114 µL of TiO${}_{2}$ NP at 15 mg/mL), the second nebulization (exp2) is not shown because of data backup failure (see Table 1).

**Figure 7 nanomaterials-12-01362-f007:**
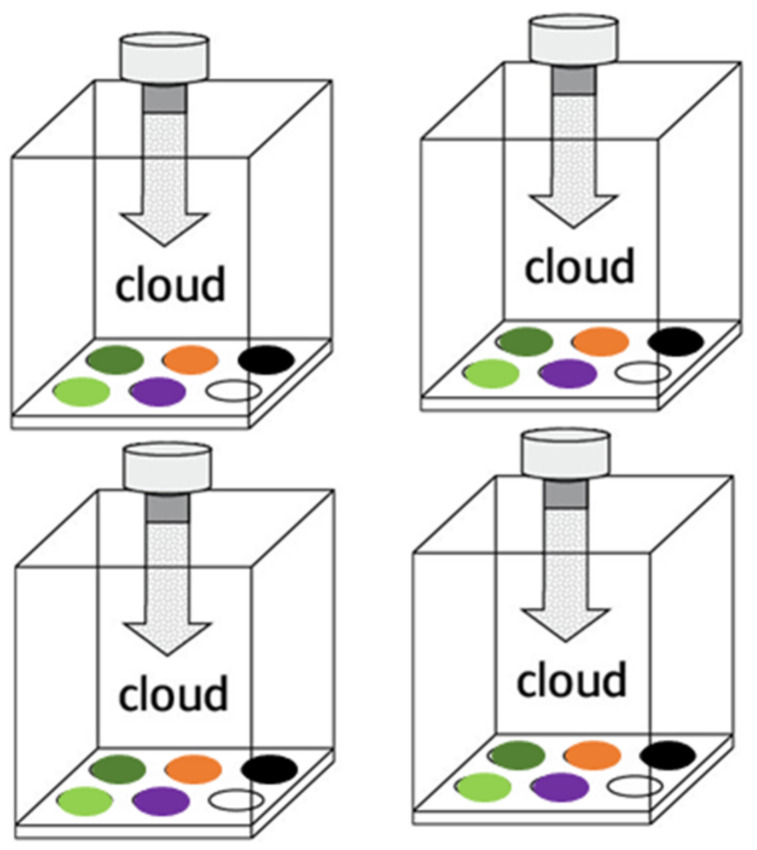
Methodology of NP exposure. One insert of each BR is placed in a well of the VITROCELL^®^ Cloud System for each of the NP exposures. After pooling of all TR, this ensures that all BR received an identical NP dose.

**Table 1 nanomaterials-12-01362-t001:** TiO${}_{2}$ NP mass deposition comparison between the initial exposure protocol and the current protocol (asymptotic values at the end are reported). Exp1, exp2, exp3 and exp4 are abbreviations for the replicates of the exposures.

	Initital Protcol (57 μL, 30 mg/mL NP)		Current Protcol (114 μL, 15 mg/mL NP)	
Exposure	Mass Deposited (μg/cm^2^)	Time (min)	Mass Deposited (μg/cm^2^)	Time (min)
Exp 1	2.1	6	4.2	5
Exp 2	7.4	24	6.1	5
Exp 3	7.4	6	5.7	5
Exp 4	3.5	6	5.5	5
Mean	5.1	10.5	5.4	5
SD	2.7	9	0.8	0

**Table 2 nanomaterials-12-01362-t002:** Quality and quantity of RNA extracted from NR8383 cells exposed to TiO${}_{2}$ NP through nebulization following the initial or current protocol. Results in red indicate a poor RNA quality or quantity preventing the use of the sample.

		Initial Protocol (57 μL, 30 mg/mL NP)			Current Protocol (114 μL, 15 mg/mL NP)		
Exposure	Biological Replicate	RNA Quantity (μg/μL)	OD 260/280	OD 260/230	RNA Quantity (μg/μL)	OD 260/280	OD 260/230
	1	89.04	1.66	1.84	32.31	2.09	2.09
	2	7.99	2.25	0.46	55.30	2.01	1.80
H${}_{2}$O	3	121.07	2.12	1.14	69.60	2.13	2.18
	4	206.30	2.04	2.32	16.09	2.08	1.43
	5	/	/	/	48.94	2.18	1.98
	1	3.30	1.86	0.26	3.61	1.83	3.72
	2	96.96	2.03	1.81	61.96	2.14	2.16
TiO${}_{2}$ NP	3	8.01	2.48	51.25	91.57	1.99	2.05
	4	134.43	2.08	1.55	58.93	2.10	2.02
	5	/	/	/	66.30	2.06	2.17
**Mean**		83.61	2.07	1.33	50.46	2.06	2.16
**SD**		73.27	0.24	0.70	26.34	0.10	0.59

## Data Availability

Not applicable.

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
