# Peer review of "Aerosol–Cell Exposure System Applied to Semi-Adherent Cells for Aerosolization of Lung Surfactant and Nanoparticles Followed by High Quality RNA Extraction"

_nanomaterials, 2022, doi:10.3390/nano12081362_

Round 1

Reviewer 1 Report

The paper reports on the investigation of covering of semi-adherent rat alveolar macrophages cultured at the air-liquid interface (ALI) with lung surfactant through nebulization. Then, the Authors have exposed the cells to the aerosolized TiO2 NM105 nanoparticles (NPs) suspension. After cells recovering, the quantity and quality of extracted from them RNA was measured. The Authors have utilized the VITROCELL® Cloud 6 System device composed of 6 quartz crystal microbalances (QCM) and the nebulizer system. The optimization of exposure to the experimental conditions, including the concentration and volume of NPs as well as organization of the exposures for their better reproducibility and repeatability was performed.

I recommend the paper for publication after minor revision addressing the issues listed below.

  1. The Authors have described culture of NR8383 cells (at line 148, Abstract) but the assessment of the deposition of nebulized and pipetted surfactant labelled with fluorescence dye (lines 178 and 334) concerns the LA-4 murine lung epithelial cells. The appropriate explanation for the benefit of general Readership should be added.
  2. References should be prepared according to the guides given in the Instructions for the Authors in the Manuscript Preparation section.
  3. Common chemical names should be typed in lower-case fonts, e.g., penicillin, amphotericin B.
  4. Figure captions in main text and supporting information should contain all important experimental conditions (e.g., concentrations of analytes, solutions, pH).
  5. The sensitivity of QCM measurements depends strongly on the resonant frequency of the quartz crystal in the QCM device. Please, provide this frequency for the convenience of general Readership.
  6. The processes of breathing/drug effects on mitochondria immobilized on a 10 MHz QCM surface were recently studied (Biosensors and Bioelectronics, 88, 2017, 114-121). This relevant literature reference should be cited.
  7. Figure 2: it is not clear if the curves exp1 ... exp4 are shown to illustrate the reproducibility of measurements or the (cumulative) effect of consecutive exposures?
  8. There are some typographical and English errors which require corrections, e.g:

Line 186: “the cell media was first completely removed” should be “the cell media were first completely removed”;

Fig. 1 caption: “resulting into” – should be “resulting in”;

Fig. 1 caption: “128 000 times per second” – redundant information (vibration frequency of the nebulizer of 128 kHz has already been provided in the same sentence).

Author Response

We would like to thank the reviewers for their thoughtful comments and efforts towards improving our manuscript.

In the attached file, the reviewers will found a document compiling 1) the list of modifications (p1-6), 2) the PDF of the fist manuscript annoted with the modification (p7-28), and finally 3) the manuscript modified (p29-51).

  1. The Authors have described culture of NR8383 cells (at line 148, Abstract) but the assessment of the deposition of nebulized and pipetted surfactant labelled with fluorescence dye (lines 178 and 334) concerns the LA-4 murine lung epithelial cells. The appropriate explanation for the benefit of general Readership should be added.

Fluorescence experiments were used as a preliminary test to evaluate the best methodology for the pre-coating of cells with surfactant. The deposition of surfactant on the cells does not depend on the cell type. Therefore, the dosimetry data reported here for LA-4 cells also apply to any other cell type, including the NR8383 cells.  This information was added in the manuscript and the methodology for for LA-4 cells used for microscopy was modified and moved in the [2.2 Culture cell] paragraph :

Line 158 : « For fluorescence microscopy, a LA-4 murine lung epithelial cell line was cultured according to ATCC guidelines: cells were seeded at 30,000 cells/cm2 in T75 flasks and split when they reached a 80% confluency. They were cultured in 10 mL of full cell medium (a mixture of F-12K medium (Gibco), 15% FBS (Fetal bovine serum, ATCC), 1% P/S (penicillin-streptomycin, Sigma), 1% NEAA (non-essential amino acids, Gibco)) in an incubator at 37°C with saturated humidity and 5% CO2. Then, cells were seeded and observed in 35 mm μ-dishes with #1.5H glass bottom (Ibidi), cells and media covered only 1/3 of the μ-dish surface (3.5 cm2) due to the geometry of the μ-dish.» was added (new lines: 169-176).

Line 333: « The deposition of surfactant on the cells does not depend on the cell type. Therefore, the dosimetry data reported here for LA-4 cells also apply to any other cell type, including the NR8383 cells. »  was added (new line: 354).

  1. References should be prepared according to the guides given in the Instructions for the Authors in the Manuscript Preparation section.

The references style was modified according to the guidelines.

  1. Common chemical names should be typed in lower-case fonts, e.g., penicillin, amphotericin B.

Lines 136-138: « Penicillin-streptomycin, Amphotericin B, Phosphate Buffered Saline (PBS), and Fetal Bovine Serum » was replaced by  « penicillin-streptomycin (PS), amphotericin B, phosphate buffer saline (PBS), and fetal bovine serum (FBS) » (new lines: 147-148).

  1. Figure captions in main text and supporting information should contain all important experimental conditions (e.g., concentrations of analytes, solutions, pH).

Figure captions were corrected and completed:

Figure 1: « resulting into » was replaced by « resulting in ».

Figure 1: « that vibrates at 128 000 times per second » was removed.

Figure 2: « This Figure illustrates the reproducibility of the QCM response for independent exposures under the same conditions. 61 μL of whole porcine surfactant solution at 40 mg/mL was nebulized, and the VITROCELLR Cloud temperature was maintained at 37°C. »  was added.

Figure 2: «  independent »  was added.

Figure 4: « TiO2 NP are nebulized over the wells of the VITROCELL Cloud 6 System containing the NR8383 cells. » was replaced by « For each 4 biological replicate (BR), NR8383 were seeded in 5 wells, in Transwell® inserts. The next day, for each BR, the inserts containing the cells were exposed, by nebulization, back-to-back to 61 µL of whole porcine surfactant at 40 mg/mL, and to 57µL of TiO2 NP at 30 mg/mL. After exposures, the Transwell® inserts containing the cells were kept at 37°C during 4 h, and the 5 wells (representing technical replicates, TR) of each BR were pooled for RNA extractions.».

Figure 5: « TiO2 NP are nebulized over the wells of the VITROCELL Cloud 6 System containing the NR8383 cells. »  « For each 5 biological replicate (BR), NR8383 were seeded in 4 wells, in Transwell® inserts, distributed in 4 6 well plates. The next day, for all 6 well plates, the inserts containing the cells were exposed (representing one technical replicate (TR) of each BR), by nebulization, first to 61 µL of whole porcine surfactant at 40 mg/mL, and then to 57µL of TiO2 NP at 30 mg/mL. After exposures, the Transwell® inserts containing the cells were kept at 37°C during 4 h, and the 4 wells (TR) of each BR were pooled for RNA extractions.».

Figure 6: « (57µL of TiO2 NP at 30 mg/mL) » and « (114µL of TiO2 NP at 15 mg/mL) » were added.

  1. The sensitivity of QCM measurements depends strongly on the resonant frequency of the quartz crystal in the QCM device. Please, provide this frequency for the convenience of general Readership.

The resonance frequency of our QCM is 5 MHz.

This information and some more technical details of the QCM were added in the manuscript :

Line 171: « The QCMs used in the VITROCELLR Cloud systems was well described previously in Ding et al. 2020 [48]. Briefly, the QCM incorporated in the VITROCELL® Cloud 6 system has an eigenfrequency of 5 MHz, a resistance of 10 Ohm and an aerosol exposed area of 4 cm², which is close to the cell covered area in 6-well Transwell® inserts (4.2 cm²).» was added (new lines: 189-193).

  1. The processes of breathing/drug effects on mitochondria immobilized on a 10 MHz QCM surface were recently studied (Biosensors and Bioelectronics, 88, 2017, 114-121). This relevant literature reference should be cited.

A paragraph concerning QCM and their applications was added and the reference suggested by the reviewer was included in cited references.

Lines 89-94:  «Quartz crystal microbalances (QCM) have been suggested for real-time measurement of the cell delivered NP dose, but its limited stability over extended periods of time have made them particularly useful for high dose rate applications (i.e. exposure times between a few minutes to hours) such as the Vitrocell®Cloud System (commercial version of ALICE Cloud system). » was replaced by « QCM are piezoelectric biosensors that detect resonance frequency variation of quartz crystal associated with mass change on their surface. During the last decade, numerous studies have reported new possible applications of QCM in biological and medical sciences including immunosensors, biosensors, microbial detectors and innovations [40–47]. Quartz crystal microbalances (QCM) have been also described as a precise device for real-time measurement of the dose-delivered in nanotoxicology studies [26,36,48]. For these reasons, a QCM has been included in the VITROCELLR Cloud System (commercial version of ALICE Cloud system) in order to control the real-time dose-delivered during the Cloud exposure.» (new lines: 95-103).

  1. Figure 2: it is not clear if the curves exp1 ... exp4 are shown to illustrate the reproducibility of measurements or the (cumulative) effect of consecutive exposures?

The four experiments (exp1… exp4) illustrate the reproducibility of measurements, the information was added in the caption of Figure 2. (see question 4)

  1. There are some typographical and English errors which require corrections, e.g:

Line 186: “the cell media was first completely removed” should be “the cell media were first completely removed”;

Fig. 1 caption: “resulting into” – should be “resulting in”;

Fig. 1 caption: “128 000 times per second” – redundant information (vibration frequency of the nebulizer of 128 kHz has already been provided in the same sentence).

The cited errors were corrected, further typos and English language were revised throughout the entire manuscript. (see the attached document “list of modifications”)

Reviewer 2 Report

The article entitled Aerosol-cell exposure system applied to semi-adherent cells for

aerosolization of lung surfactant and nanoparticles followed by

high quality RNA extraction is a document of interesting subject matter.

However, it needs some major changes before being accepted. Make the following corrections:

  1. In the Introduction, the authors need to elaborate on the role of nanoparticles as theranostic tools in the fight against different cancers and diseases by citing and briefly discussing the following papers (DOI: 10.3390/nano11113002, DOI: 10.3390/pr9040621,DOI: 10.1007/s40089-021-00332-2, DOI: 10.2174/0929867328666210810160901, DOI: 10.3390/nano11102579, DOI: 10.1016/j.jksus.2021.101724 ).
  2. It is expected to have an extensive literature review followed by an in-depth and critical analysis of the state of the art, and identify challenges for future research.
  3. Pay attention on the more interpretation of the experimental results and doing comparison with former reports in this field.
  4. Your abstract should clearly state the essence of the problem you are addressing, what you did and what you found and recommend. That will help a prospective reader of the abstract to decide if they wish to read the entire article.
  5. The conclusion section can be refined better. Please indicates if ‎your objectives were ‎reached, in what your work is novel and confirms or not, previous findings. Also, ‎‎some perspectives generally arise from your investigations and must be indicated here. ‎‎
  6. Generally, there are numerous typos in the manuscript that have to be corrected.
  7. The objective or objectives should be clearly elucidated in the last paragraph of the introduction.

Author Response

We would like to thank the reviewers for their thoughtful comments and efforts towards improving our manuscript.

In the attached file, the reviewers will found a document compiling 1) the list of modifications (p1-6), 2) the PDF of the fist manuscript annoted with the modifications (p7-28), and finally, 3) the manuscript modified (p29-51).

  1. In the Introduction, the authors need to elaborate on the role of nanoparticles as theranostic tools in the fight against different cancers and diseases by citing and briefly discussing the following papers (DOI: 10.3390/nano11113002, DOI: 10.3390/pr9040621,DOI: 10.1007/s40089-021-00332-2, DOI: 10.2174/0929867328666210810160901, DOI: 10.3390/nano11102579, DOI: 10.1016/j.jksus.2021.101724 ).

A paragraph including these references and some others was added in the introduction:

Line 27: « Moreover, the emergence of nanotechnologies has led to the development of new research fields such as nanomedicine, which focuses on the synthesis and engineering of nanomaterials for drugs delivery [4], which has led to novel treatment [5–8] and diagnostic options [9,10] and combinations thereof [11,12]. Consequently, the study of both adverse and therapeutic health effects of nanoparticles and new nanomaterials is a crucial issue for the development of safe nano-enabled products such as consumer products as well as medical products.» was added (new lines: 27-33).

  1. It is expected to have an extensive literature review followed by an in-depth and critical analysis of the state of the art, and identify challenges for future research.

We updated in the manuscript with the last published references in regard with the topic and the text was modified as well (introduction, conclusion).

  1. Pay attention on the more interpretation of the experimental results and doing comparison with former reports in this field.

As far as we know, we are the first team to publish this kind of very specific report, dedicated to transcriptomic studies. That’s why, we focused on the comparison between the pre-established protocol and our optimized protocol. Nevertheless, some references were added and discussed.

  1. Your abstract should clearly state the essence of the problem you are addressing, what you did and what you found and recommend. That will help a prospective reader of the abstract to decide if they wish to read the entire article.

To clarify our statement, the abstract was modified as follows:

Lines 4-15: « The commercially available VITROCELL® Cloud System has been applied for the delivery of aerosolized substances to adherent cells under ALI conditions, but not yet to semi-adherent cells and surfactant. In this work, semi-adherent rat alveolar macrophages NR8383 grown at the ALI were coated with lung surfactant through nebulization using the VITROCELLR Cloud 6 System before being exposed to TiO2 NM105 nanoparticles (NPs). The nebulization of surfactant forms a homogeneous layer over all the cells, reproducing alveolar physiological conditions. After NP exposures, RNA was extracted and its quantity and quality were measured. The results highlighted the need for adaptation of the standard operating procedure, especially concerning physiologic surfactant conditions as well as reproducibility and repeatability of NP mass deposition and RNA quality and quantity. This led us to adapt the standard operating protocol of the VITROCELL® Cloud System and to present our recommendations for in vitro ALI NP aerosolization of semi-adherent cells pre-coated with lung surfactant » was replaced by «  While the commercially available VITROCELL® Cloud System has been applied for the delivery of aerosolized substances to adherent cells under ALI conditions, it has not yet been tested on lung surfactant and semi-adherent cells such as alveolar macrophages, which are playing a pivotal role in the nanoparticle-induced immune response. Objectives : In this work, we developed a comprehensive methodology for coating semi-adherent lung cells cultured at the ALI with aerosolized surfactant and subsequent dose-controlled exposure to nanoparticles  (NPs). This protocol is optimized for subsequent transcriptomic studies. Methods: Semi-adherent rat alveolar macrophages NR8383 were grown at the ALI and coated with lung surfactant through nebulization using the VITROCELLR Cloud 6 System before being exposed to TiO2 NM105 NPs. After NP exposures, RNA was extracted and its quantity and quality were measured. Results: The VITROCELL® Cloud system allowed for uniform and ultrathin coating of cells with aerosolized surfactant mimicking physiological conditions in the lung. While nebulization of 57 μL of 30 mg/mL TiO2 and 114 μL of 15 mg/mL TiO2 nanoparticles yielded identical cell delivered dose, the reproducibility of dose as well as the quality of RNA extracted were better for 114 μL.» (new lines: 4-17).

  1. The conclusion section can be refined better. Please indicates if ‎your objectives were ‎reached, in what your work is novel and confirms or not, previous findings. Also, ‎‎some perspectives generally arise from your investigations and must be indicated here.

To clarify our statement, the conclusion was modified as follows:

Lines 560-568: « The total RNA extracted from this VITROCELLR Cloud exposition was used to perform a complete transcriptomic study [43] including a comparison of in vivo and air-liquid interface and submerged in vitro results for TiO2 NM105 NP exposure [42]. Owing to the specific objectives of this study, the recommended experimental procedure for operating the VITROCELLR Cloud 6 described suggests using lower nebulization volumes for the surfactant (61 μL) and the TiO2 NP suspension (114 μL) as compared to the manufacturer-recommended 200 μL to keep the cell-deposited liquid layer as low as possible. However, the total delivered amount of liquid of 175 μL is relatively close to the 200 μL recommended by the manufacturer. To conclude, » was replaced by « The goal of this work was to refine an ALI pre-established protocol. As far we know, such a protocol has not been previously published. Therefore, after addressing each crucial steps, (i.e.. the use of very low concentrations of NP, the deposition of a homogeneous layer of surfactant, the specific use of semi-adherent cells, and finally the enhancement of the extraction yield of RNA to carry out transcriptomic studies), we present here for the first time a robust almost ready-to-use protocol. This one will be helpful for nanotoxicologists interested in developing such methodologies. This protocol is not frozen, and researchers can feel free to suggest improvements. Finally, » (new lines: 583-591).

Lines 596-599: «  Finally, considering the improvement of co-culture or 3D-cultures techniques, coupled with an innovative exposure device such as the VITROCELL® Cloud, it is reasonable to think that in the future this kind of methodology will contribute to reducing the use of animals in toxicology research. » was replaced by « After the optimization of this protocol, we were able to obtain a high quality and quantity of total RNA extraction from this VITROCELLR Cloud exposition and we performed a complete transcriptomic study [64] including a comparison of in vivo and air-liquid interface and submerged in vitro results for TiO2 NM105 NP exposure [63]. The optimization of the existing protocols is essential when moving to new or different experimental conditions (e.g., in our case of semi-adherent cell cultures, surfactant and low doses of NPs). The method that we employed to optimize/refine the previously established protocol could efficiently be transposed to 3D-cultures [87–90] or culture of organoids [91–95], which represent today the in vitro models closest to physiological conditions, considering the interactions existing between the different cell types. Furthermore, the approach that we used could serve as a basis for further developments which, in the long term, could significantly reduce the need of animal studies for toxicology research. » (new lines: 619-631).

  1. Generally, there are numerous typos in the manuscript that have to be corrected.

The cited errors were corrected, further typos and English language were revised throughout the entire manuscript. (see attached document “list of modifications”)

  1. The objective or objectives should be clearly elucidated in the last paragraph of the introduction.

To clarify our statement, the final paragraph of the introduction was modified as followed:

Lines 126-132: « Here, we present a methodology for pre-coating semi-adherent cells cultured at ALI with aerosolized surfactant immediately before NP exposure through the VITROCELL® Cloud 6 System and subsequent extraction of their RNA. This optimization was done to provide high quality and sufficiently efficient RNA extraction as the first step of transcriptomic analysis of cells exposed to NP in the VITROCELL® Cloud System (article previously published, Leroux M.M. et al., 2020 [42,43]), which provides important insight into the toxicological profile of NPs.» was replaced by « To sum up, in vitro studies based on ALI exposure are more and more used in toxicology studies. But researchers using such a strategy have to face some issues including 1) using a very low concentration of NP 2) applying a homogeneous layer of surfactant on cells 3) using semi-adherent cells in ALI, and 4) retrieving RNA in sufficient quantity and quality to carry out transcriptomic studies. Thus, here, we present an in-depth detailed protocol answering to these points, in which each steps are validated, after the optimization of the different parameters. For a better understanding, all experimental results are featured in the present paper. It could be of help for researchers using such a device.» (new lines: 135-143).

Round 2

Reviewer 2 Report

Authors addressed all comments carefully. 

My suggestion is "Acceptance " now.